# From Tradition to Innovation: Analyzing Strategies and Support for Enhancing Badminton Course Teaching Quality through Educational Technology

**DOI:** 10.3390/bs14090857

**Published:** 2024-09-23

**Authors:** Mei-Ling Lin, Nan-Chi Chen, Yu-Jy Luo, Chun-Chin Liao, Chun-Chieh Kao

**Affiliations:** 1General Education Center, MacKay Junior College of Medicine, Nursing, and Management, Taipei 112021, Taiwan; s091@mail.mkc.edu.tw; 2Department of Tourism and Leisure, Da-Yeh University, Changhua City 515006, Taiwan; n5477@mail.dyu.edu.tw; 3Office of Physical Education, Ming Chuan University, Taoyuan City 333321, Taiwan; luck@mail.mcu.edu.tw

**Keywords:** multimedia-assisted teaching, physical education teaching quality, teaching effectiveness

## Abstract

This study aims to explore the impact of Multimedia-Assisted Teaching (MAT) on the quality of physical education (PE) instruction in universities, with a particular focus on badminton courses. A quasi-experimental design was employed, including pre-tests and post-tests, involving two badminton classes at Ming Chuan University, with a total of 101 students. These two classes were assigned to an experimental group (using MAT) and a control group (traditional teaching methods). The research instrument used was the Physical Education Teaching Quality Scale (PETQ), which includes four factors: course content, teaching strategies, classroom management, and learning assessment. The results indicated that, compared to traditional teaching methods, the use of MAT significantly enhanced students’ perception of the quality of PE instruction. The experimental group scored significantly higher than the control group across all scale factors, suggesting that MAT is more effective in capturing students’ attention and improving learning outcomes. The conclusion suggests that MAT has significant advantages in improving the quality of physical education instruction. The integration of MAT enables more flexible lesson planning and enhances the learning process through the features of multimedia-assisted teaching. It is recommended that physical education teachers consider applying multimedia technologies to aid teaching, in order to increase student engagement and effectiveness.

## 1. Introduction

### 1.1. Research Background

In recent years, the rise of social media websites has further accelerated the dissemination of information traditionally conveyed through media. The interplay between these two mediums has generated explosive growth, making multimedia an indispensable tool in everyday life. However, in the context of physical education (PE), the role of multimedia remains underexplored. This study focuses on how multimedia can be applied effectively in PE courses, specifically badminton, to enhance learning outcomes and teaching quality. Teachers can leverage the advantages of multimedia by adopting a multifaceted approach to utilizing various multimedia resources to create real-time and interactive learning environments. This approach ensures that the educational environment supports students in maintaining a positive attitude and strong intrinsic motivation [1]. Multimedia, incorporating various forms such as symbols, images, graphics, audio, video, and animations, often leverages technology to improve understanding and memory retention [2]. Given the challenges of teaching complex motor skills in PE, multimedia provides a practical solution for making abstract concepts more understandable. Educators should adapt to these diverse media formats, as both teachers and learners aim to achieve the key educational goals of comprehending and retaining the material’s meaning [3].

The use of multimedia in blended learning environments can significantly enhance the overall learning experience. As cognitive processes during media-based learning have become a focus of recent research [4], multimedia content such as videos, animations, and interactive simulations can deeply engage students, making the learning process more dynamic and captivating. Research [5] shows that multimedia tools better explain badminton rules, techniques, and strategies compared to traditional methods, enhancing student performance and understanding. This sustained engagement helps maintain student interest in the material and promotes active participation, which in turn improves retention and leads to better learning outcomes [6]. With its ability to combine various forms of media, multimedia tools have become essential in education. Through real-time interaction and enriched content, teachers can effectively motivate students, improve understanding, and ultimately achieve superior educational results. Additionally, multimedia allows for differentiated learning, catering to diverse student needs and preferences.

PE is the only setting in which all students experience instructed physical activity. In order to meet students’ needs and motives to be physically active, PE ideally provides various movement experiences covering different strands, which are presented with different emphases [7]. PE is one of the few subjects in educational development that can provide students with balanced growth across the three domains of skills, affect, and cognition [8]. PE focuses on physical activities and encompasses both educational and sports science fields. Through movement, games, and sports, it interprets and inspires human potential and creativity, achieving the goal of holistic education. Previous studies have found [9,10,11] that multimedia instructional videos effectively capture students’ attention and enhance understanding by providing clear, detailed demonstrations of complex movements. This approach complements traditional teacher-led instruction, offering visual aids that facilitate pre-class preparation, in-class demonstrations, real-time feedback, and post-class review. Reasonable use of multimedia teaching in PE classes can transform complex, abstract, and difficult-to-learn motor skills into concrete, understandable knowledge for students. This helps reinforce their impressions, acquire skills and experience, and enable meaningful learning, thus facilitating the achievement of educational objectives [12]. It can also significantly enhance students’ understanding of the subject and improve the quality of education [13]. Moreover, according to research by Cheng [14], by using new educational concepts, methods, and means, with the rapid development of modern educational technology, multimedia technology, and network technology and the combination of the three, network multimedia courseware will become one of the main resources for teaching in the future. Using the network, teachers and students can realize personalized teaching and learning at different times and in different places and, at the same time, provide a realistic basis for a variety of learning styles. In other words, an effective PE curriculum not only provides positive learning experiences, enabling students to develop essential motor skills needed throughout their lives, but also encourages students to analyze, reflect upon, communicate, and apply these skills [15]. PE offers balanced growth across skills, affect, and cognition. Utilizing multimedia instructional videos in PE can transform complex motor skills into comprehensible knowledge, enhancing student engagement and learning outcomes. Therefore, effective PE curricula also promote lifelong exercise habits and essential motor skills development.

Teaching is a series of dynamic processes, making comprehensive evaluation challenging [16]. The process and outcomes of teaching are crucial factors affecting the quality of education [17]. The achievement rate of teaching objectives is often used as a measure of teaching quality. In this context, the study will measure the effectiveness of multimedia integration by assessing how well it helps achieve the teaching objectives in badminton courses. However, these objectives vary with educational purposes and the context of the times, leading to different demands on teaching [12]. Classroom teaching includes various factors such as teaching conditions, curriculum difficulty, and learning outcomes [18]. By incorporating multimedia into these variables, this research aims to determine its impact on student engagement and skill development. Due to differences in academic fields and explored dimensions, Boonsem and Chaoensupmanee [18] identified six aspects affecting the effectiveness of physical education teaching: (a) the purpose of physical education learning; (b) the content and curriculum of physical education; (c) teaching methods and activities; (d) the personality of the physical education teacher; (e) equipment and facilities; and (f) assessment and evaluation. Additionally, Liu et al. [19] specifically pointed out that the quality scale of university physical education teaching includes four stable factors: (a) curriculum content; (b) teaching strategies; (c) classroom management; and (d) learning assessment. These factors are used to examine whether the teaching content provided by teachers meets students’ needs and expectations. Evaluating the significant value of physical education courses relies on the appropriate teaching strategies employed by physical education teachers to inspire students’ interest in sports, stimulate their learning motivation, and enhance learning outcomes [20]. Evidently, assessing university teachers’ classroom teaching and quality is essential for promoting teaching improvement and enhancing educational quality. Improving the teaching quality of professional teachers through teaching strategies can elevate students’ learning motivation and effectiveness.

Reflecting on the trajectory of online teaching during the pandemic, some scholars believe that multimedia learning will become mainstream in the future. However, concerns have been raised regarding the negative impacts of online teaching. Whether distance or multimedia-assisted teaching supplements or replacing traditional methods is a topic worth considering and exploring. Physical education teachers often face certain challenges when using this technology, including a lack of time, insufficient expertise, and limited access to resources. It is important to recognize that integrating multimedia information technology into teaching may increase the workload for teachers. Additionally, excessive sensory stimulation may lead to cognitive overload for users, causing confusion and reducing the effectiveness of message transmission. Effectively utilizing existing technology and resources to stimulate students’ visual interest through multimedia, thereby enhancing their perception of the quality of physical education instruction, is one of the primary motivations of this study. The second motivation is the limited research on the teaching quality of university badminton courses. Therefore, the purpose of this study is to create a multimedia learning environment for badminton courses and to conduct a four-week experimental intervention using a quasi-experimental design with pre- and post-tests. The intervention includes instructional materials on aspects such as court equipment, game rules, and scoring methods. The study uses the Physical Education Teaching Quality Scale (PETQ) as a research tool to explore the impact of traditional learning environments versus multimedia learning environments on the quality of physical education teaching.

### 1.2. Research Hypotheses

The purpose of this study is to create a multimedia learning environment for PE classes and to determine the impact of both traditional learning environments and multimedia learning environments on the quality of PE instruction. Specifically, we have the following objectives. First, we aim to analyze the effects of traditional teaching (TT) and MAT on the quality of PE instruction. Second, our objective is to compare the differences in students’ perceived quality of PE instruction between TT and MAT. Based on the aforementioned influencing factors and a review of the literature, this study proposes that MAT can enhance the quality of PE instruction for students. Therefore, we propose the following research hypothesis:

**Hypothesis 1:** 
*Students’ perceived quality of PE instruction significantly improves under MAT, and this improvement is more pronounced compared to students receiving traditional teaching methods (TT).*


## 2. Methods

### 2.1. Study Sample

We enrolled 101 students (As shown in Figure 1) from two badminton classes at Ming Chuan University in Taiwan. The included participants were divided into the Experimental group (*n* = 52; age: 19.92 ± 0.97 years; male: 26, female: 26) and the direct instruction (control) group (*n* = 49; age: 19.86 ± 1.12 years; male: 29, female: 20) for class-based experimental instruction.

### 2.2. Experimental Procedures

Because of limitations in the educational environment and class size of the original classroom setting, we were unable to conduct a randomized controlled trial or other multifactor true experimental design. Instead, a quasi-experimental design with an unequal sample-size pretest–posttest design was employed for a 4-week experimental intervention. The experimental venue is the badminton court of Physical Education Hall 1, Taiwan Ming Chuan University, Taoyuan Campus. The experimental procedures of the study included the pretest, intervention, and posttest stages. First, the students were recruited and screened based on inclusion and exclusion criteria. The two classes were assigned to be the experimental or the direct instruction group. The course structure was explained to the students, and all participants signed a consent form. A physical education class in the participating schools lasted 100 min once a week for 4 weeks. Based on the team’s developed materials covering aspects such as court equipment, game rules, and scoring methods Q&A (each unit approximately 4 min), each week after oral instruction, students were immediately asked to use their mobile phones for video-assisted teaching. The intervention courses were taught by certified physical education teachers with over 22 years of teaching experience. Both groups received the same physical education content and skills instruction. The study protocol was approved by the Institutional Review Board of National Taiwan University before study commencement (Case number: 202204EM020).

### 2.3. Research Instruments

The research instrument utilized in this study was the Physical Education Teaching Quality Scale (PETQ), developed by Liu et al. [19]. It consists of 20 items, encompassing four factors: “Curriculum Content (CC)”, “Teaching Strategies (TS)”, “Classroom Management(CM)”, and “Learning Assessment (LA)”. The scale employs a five-point Likert scale, ranging from “Strongly Agree” to “Strongly Disagree,” scored as 5, 4, 3, 2, and 1, respectively. Higher scores indicate higher perceived quality of physical education teaching. The overall fit indices of the Physical Education Teaching Quality Scale are as follows: the absolute fit index, χ^2^/df = 2.16, which is less than 3 (χ^2^ = 353.66; df = 164); GFI = 0.86, which did not reach 0.90; SRMR = 0.03, less than 0.05; RMSEA = 0.07, less than 0.08 (90% CI = 0.062~0.082). The standardized coefficients of composite reliability ranged from 0.66 to 0.84. In terms of construct reliability and average extracted variance, the composite reliability values of the four factors ranged from 0.88 to 0.90. The discriminant validity test yielded ∆χ^2^ values exceeding 3.84 for all factors. Overall, most of the required acceptance values were met, indicating that the model is acceptable. The Physical Education Teaching Quality Scale developed in this study demonstrates good reliability and validity, with discriminant effects, suggesting consistency and stability of the scale.

### 2.4. Curriculum Development

The research team, composed of two professional badminton referees and one multimedia teacher, based their development on the rules published by the Badminton World Federation (BWF). Following three focus group meetings, the team determined the direction for developing teaching materials. Subsequently, based on the conclusions drawn from these meetings, the team developed materials covering aspects such as court equipment, game rules, and scoring methods Q&A. As shown in Figure 2, the video presentations were delivered in Traditional Chinese and presented in MP4 format (The videos used were created and uploaded by the authors, ensuring full copyright and legal rights).

### 2.5. Statistical Analysis

Statistical analysis was conducted using the SPSS 22.0. The results were analyzed using five methods; (1) participants’ height, weight, and body mass index (BMI) were analyzed using descriptive statistics; (2) homogeneity in sex was tested using a chi-squared test; (3) homogeneity in the participants’ height, weight, and BMI was tested using independent sample *t*-tests; (4) after the effect of pretest scores was eliminated, the effect of the course on the physical education teaching quality of the experimental and control groups was analyzed using analysis of covariance; (5) the effect size of the intervention was calculated; and (6) the significance level for all statistical tests in this study was set at α < 0.05.

## 3. Results

### 3.1. Homogeneity Testing

Analysis of the demographic statistics revealed homogeneity between the groups in terms of gender (χ^2^ = 0.824, *p* > 0.05), age (t = 0.318, *p* > 0.05), height (t = 1.605, *p* > 0.05), weight (t = 0.106, *p* > 0.05), and BMI (t = 0.866, *p* > 0.05). This indicates that the experimental and control groups exhibit homogeneity in individual characteristic variables. The participants’ demographic statistics are listed in Table 1.

The homogeneity of the within-class regression coefficients for the PETQ (F = 308.04; *p* = 0.063, η^2^ = 0.059) variables was consistent with the hypothesis regarding the homogeneity of the within-class regression coefficient in the covariance analysis (Table 2). The data had high homogeneity and were ideal for experimental intervention. Given that the results supported the hypothesis, the data were suitable for covariance analysis.

### 3.2. Comparison of the Performance of Physical Education Teaching Quality between the Experimental Group and the Control Group

As indicated in Table 3, the scores of students in the experimental group for CC, TS, CM, LA, and the overall PETQ scale were 4.59 (ADJ Mean = 4.59), 4.62 (ADJ Mean = 4.60), 4.64 (ADJ Mean = 4.66), and 4.63 (ADJ Mean = 3.63), respectively. The overall scale of PETQ for this group was 4.67 (ADJ Mean = 4.67). The scores of students in the control group for CC, TS, CM, LA, and PETQ were 3.63 (ADJ Mean = 3.62), 3.71 (ADJ Mean = 3.70), 3.71 (ADJ Mean = 3.69), and 3.69 (ADJ Mean = 3.68), respectively. The scale of PETQ for this group was 3.74 (ADJ Mean = 3.71).

Based on the data from Table 3 and Table 4, subsequent to excluding the influence of the covariate (pretest scores) on the dependent variable (post-test scores), ANCOVA (Analysis of Covariance) results indicated remarkable outcomes in terms of PETQ due to the implementation of 4 weeks’ multimedia audio-visual teaching materials in the experimental group. Specifically, post-test scores of the experimental group in CC (M = 4.59 > 3.62), TS (M = 4.60 > 3.70), CM (M = 4.66 > 3.69), LA (M = 4.63 > 3.68), and the overall PETQ scale (M = 4.67 > 3.71) were notably superior to those of the control group. Moreover, F values of 39.27, 20.91, 16.44, 121.35, and 37.33 were observed, indicating statistical significance (*p* < 0.05). According to the η^2^ (eta squared) standard proposed by Cohen [21], the effect size results of physical education teaching quality obtained in this study had a medium effect; the η^2^ values ranged from 0.15 to 0.30. The experimental course developed in this study significantly improved the physical education teaching quality of participants. These results suggest that students in the experimental group improved their PETQ more than their peers in the control condition.

## 4. Discussion and Study Limitations

### 4.1. Discussion

This study investigates the impact of MAT on the quality of PE classes. The results suggest that a four-week badminton MAT program effectively enhances students’ perception of PE class quality, including aspects such as curriculum content, teaching strategies, classroom management, and learning assessment. These findings align with previous research [10,22,23], which also highlights that the integration of multimedia-assisted teaching in badminton PE courses can significantly promote positive effects on the quality of physical education teaching. Furthermore, the results indirectly support the viewpoints of Faraday and Sutcliffe [24] and Zheng et al. [25], suggesting that the appropriate combination of various media and teaching methods can enhance information comprehension. In the era of online multimedia, physical education, like other subjects, can utilize online media resources to enrich course content, offering an opportunity to efficiently enhance the quality of physical education instruction. This study also corroborates the research directions of Rumiantseva et al. [12], Zhou [26], and Liang et al. [27], showing that integrating innovative technologies into the educational process can significantly improve teaching quality. Multimedia possesses unique qualities that traditional teaching methods cannot replace, such as interactivity and intuitiveness. When these qualities are applied to physical education classes, they positively impact students’ learning beliefs, learning environment, and learning outcomes, further promoting changes in learning methods and providing a positive learning experience for physical education classes. However, it should not be overlooked that traditional teaching methods also have unique qualities that multimedia cannot replace, such as real-time interaction and emotional exchange between teachers and students, as well as group cooperation and in-person teaching activities. These are aspects that multimedia-assisted teaching finds difficult to fully replicate. Vagg et al. [28] remind us that while multimedia can serve as an essential and efficient supplementary learning tool to enhance traditional teaching methods, it cannot replace them entirely. The interaction between teachers and students, the cultivation of practical skills, the development of social and cooperative skills, and the diversity of teaching content and methods in PE classes still require the support and supplementation of traditional teaching methods. The optimal teaching strategy should be an efficient combination of multimedia technology and traditional teaching methods, leveraging their respective advantages to achieve the best teaching outcomes.

With the declining birth rate in Taiwan, parents are increasingly concerned about the quality of education their children receive, leading to heightened expectations and demands for school performance and educational quality. Physical education, as a component of education, is no exception. Therefore, using the concept of service quality to enhance the teaching quality of physical education and to attract and meet the needs of students is a critical issue that PE teaching units must address [19]. PE teaching units should design and provide physical education programs based on student’s interests, preferences, and expectations to ensure active student participation and meet their learning and developmental needs. Li and Liu [29] confirmed this in their study, where they found a significant 25% improvement in student satisfaction and a 30% increase in learning efficiency, highlighting the potential of multimedia platforms to transform physical education. Additionally, multimedia technology also bridges the gap in providing quality educational opportunities and enhances learners’ performance [30]. Thus, improving teaching quality is crucial for attracting students and meeting their diverse needs. There is a direct causal relationship between the quality of classroom instruction and student learning achievements [31], which also affects student satisfaction [9], learning motivation [32], and academic performance [33]. In physical education classes, limited curriculum and large student numbers often hinder the teaching process. The integration of MAT into physical education offers an opportunity to more effectively address these issues without hindering student learning. That is to say, the combination of video materials with lesson plans can effectively guide students to master key movements and techniques, thereby enhancing teaching quality and improving student learning outcomes.

In terms of curriculum, MAT involves sharing learning materials before class and engaging in active discussions during class [34,35]. In a multimedia learning environment, learners interact with visual and auditory information, which is processed into short-term memory and then encoded into long-term memory [36]. The combination of MAT and TT methods, with TT as the primary approach and MAT as a supplementary tool, extends the learning scope and shows significant effects [11]. However, findings that suggest MAT better captures student attention, enhances performance, and provides more effective assessments [37]. This should be interpreted with caution, as many of these conclusions are based on self-report methods. These methods, while useful, have inherent limitations, such as social desirability bias and recall inaccuracies, which may affect the reliability of the data. By integrating the teacher’s teaching philosophy and curriculum content with multimedia elements such as text, graphics, images, animations, sound, and video, MAT presents lively and diverse content, fostering student self-management and learning abilities, and bridging the gap between teachers and students. This approach allows teachers to diversify learning methods within the educational environment, thereby enhancing teaching quality and effectiveness. In other words, using multimedia tools to develop a more diverse teaching model, integrating online and offline elements, might be a valuable teaching strategy. From the teacher’s perspective, MAT promotes active learning, emphasizing student-initiated learning [38]. While it helps improve the utilization and efficiency of teaching resources, it is important to note that some of the reported benefits of MAT, such as time-saving or enhanced student engagement, may also be influenced by the subjective nature of self-reported feedback. Therefore, care should be taken when generalizing these outcomes. Media technology’s integrative, diverse, and interactive features enable teachers to use digital and print elements to communicate information and ideas more effectively, facilitating better concept comprehension and information transmission to learners [39]. In TT, teachers often spend considerable time preparing teaching materials, whereas MAT, through electronic lesson plans, teaching videos, and online courseware, allows for quick access to and presentation of rich teaching resources, saving preparation time.

In conclusion, with the continuous advancement of technology, the application of multimedia-assisted teaching (MAT) will become increasingly widespread and significantly promote the improvement of educational quality. The results indicate that the use of MAT profoundly impacts teaching quality. Its main advantage lies in the ability to integrate various media forms such as text, images, audio, video, and animations, providing rich learning resources and interactive experiences that enhance students’ interest and participation. MAT’s teaching content is more vivid and intuitive, whereas traditional teaching (TT) mainly relies on verbal explanations and technical demonstrations, which sometimes fail to capture students’ attention. In contrast, multimedia can concretize abstract concepts through images, animations, and videos, making it easier for students to understand and remember. However, this does not mean that these effects cannot be achieved through traditional teaching methods (TT). In fact, some TT methods can effectively capture students’ attention, especially when the teacher excels in verbal expression and technical demonstrations. MAT’s strength lies in its multimedia presentation and interactive features, but traditional teaching can also achieve similar outcomes in different contexts. Therefore, we cannot entirely dismiss the possibility that TT may yield similar educational outcomes in certain situations. Overall, MAT’s core concept is to revolutionize traditional teaching by utilizing teacher-produced videos and interactive lessons, allowing what was once taught in the classroom to be learned at home before class. The classroom thus becomes a place for problem-solving, concept expansion, and collaborative learning. MAT holds significant advantages in enhancing teaching quality by providing intuitive, dynamic learning content, fostering personalized and collaborative learning, and improving the efficiency and effectiveness of teaching resources.

Despite the research results and literature supporting MAT as a developmental direction for physical education courses and instruction [40], physical education teachers often face common barriers when using this technology. These barriers include time constraints, professional knowledge, and access to resources [41]. It is crucial to acknowledge that integrating multimedia information technology into teaching can be a substantial burden. Additionally, excessive sensory stimulation can lead to cognitive overload for users, causing confusion and reducing the effectiveness of information dissemination [42]. Therefore, the use of multimedia in teaching must consider the cognitive load of learners. This involves incorporating cognitive load into instructional design, adjusting the amount of presented material, and appropriately integrating multimedia technology to clearly convey the content of the material [11]. Multimedia enhances learning and the quality of physical education instruction, making it an indispensable aspect of future education and learning. From an educational standpoint, it is essential to optimize learners’ cognitive resources to achieve the best learning outcomes. Establishing a connection between multimedia usage and physical education instruction involves balancing the avoidance of cognitive overload and fostering meaningful and enjoyable learning experiences. This balance represents the primary task and challenge in designing multimedia-based teaching.

### 4.2. Study Limitations

Due to time constraints and the necessity to consider factors such as educational objectives, curriculum scheduling, and teaching resources, this study only selected four teaching units suitable for university-level badminton instruction. Consequently, not all badminton teaching activities were comprehensively incorporated, requiring cautious inference. This limitation is acknowledged within the scope of this study.

## 5. Conclusions and Recommendations

This study fills a gap in the literature by examining the benefits of teaching quality in university PE classes within the context of MAT. The results indicate that the integration of MAT has a positive impact on the teaching quality of PE classes for participants, encouraging PE instructors to adopt MAT in their teaching practices. In practical terms, MAT enables PE instructors to utilize multimedia resources, thereby enhancing students’ understanding of the course and promoting more effective learning outcomes. Additionally, this study provides strong evidence of MAT’s impact on the teaching quality of university badminton courses. The findings suggest that the use of MAT can strengthen lesson planning, improve classroom management, and facilitate personalized feedback, making it easier for instructors to adapt to the diverse needs of students. These strategies can be adopted in other university PE activities to improve instructional delivery and curriculum development. Furthermore, future research is recommended to reapply this study by investigating the impact of MAT on different PE courses and various course durations.

## Figures and Tables

**Figure 1 behavsci-14-00857-f001:**
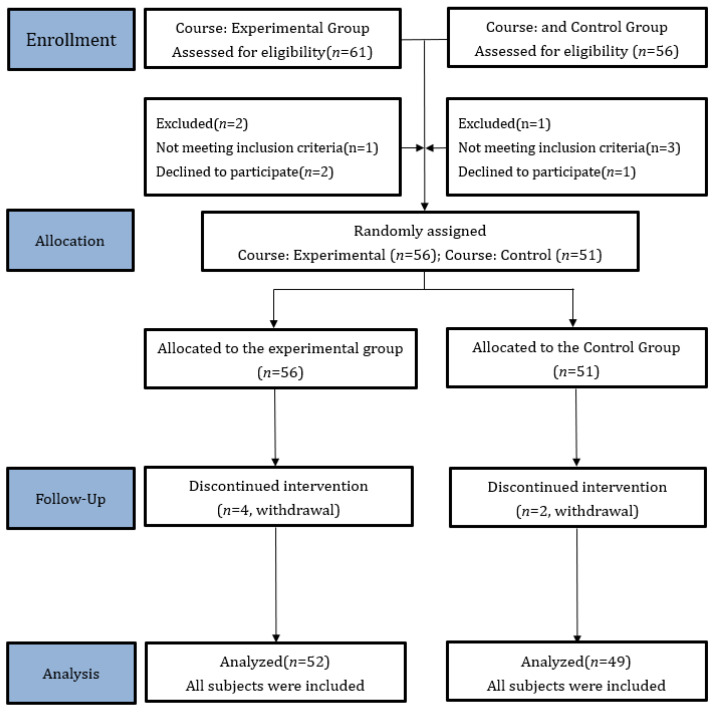
CONSORT 2010 Flow Diagram for the study.

**Figure 2 behavsci-14-00857-f002:**
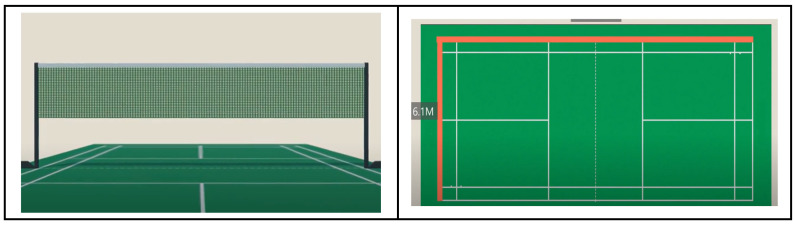
Screen capture of the multimedia videos (source: https://youtu.be/Ty-iM6Z-yoI, accessed on 14 June 2024).

**Table 1 behavsci-14-00857-t001:** Demographic statistics of participants.

Variable	Experimental (*n* = 52; *M* [*SD*])	Control Group (*n* = 49; *M* [*SD*])	Significance
Gender (male: female)	26:26	29:20	χ^2^ = 0.824, *p* > 0.05
Age (years)	19.92 [0.96]	19.86 [1.11]	t = 0.318, *p* > 0.05
Height (cm)	165.35 [8.90]	162.58 [8.38]	t = 1.605, *p* > 0.05
Weight (kg)	59.10 [13.99]	58.84 [10.30]	t = 0.106, *p* > 0.05
BMI (kg/m^2^)	21.21 [4.41]	22.15 [2.70]	t = 0.866, *p* > 0.05

*p* < 0.05.

**Table 2 behavsci-14-00857-t002:** Homogeneity of within-class regression coefficients of PETQ.

Source of Variation	Type III SS	df	MS	F	*p*	η^2^
PETQ	8.43	1	8.43	308.04	0.063	0.059
	2.68	98	0.027			

*p* < 0.05.

**Table 3 behavsci-14-00857-t003:** Descriptive statistics of PETQ in the experimental and control groups.

Group	Source	Pre-Test (SD)	Pro-Test (SD)	Adjusted Mean
Experimental	CC	3.69 (0.46)	4.59 (0.33)	4.59 (0.34)
	TS	3.71 (0.43)	4.62 (0.40)	4.60 (0.38)
	CM	3.64 (0.49)	4.64 (0.42)	4.66 (0.36)
	LA	3.70 (0.50)	4.63 (0.28)	4.63 (0.18)
	PETQ	3.69 (0.49)	4.67 (0.34)	4.67 (0.41)
Control	CC	3.77 (0.46)	3.63 (0.50)	3.62 (0.35)
	TS	3.67 (0.43)	3.71 (0.52)	3.70 (0.39)
	CM	3.69 (0.49)	3.71 (0.48)	3.69 (0.37)
	LA	3.71 (0.42)	3.69 (0.44)	3.68 (0.18)
	PETQ	3.71 (0.36)	3.74 (0.49)	3.71 (0.42)

**Table 4 behavsci-14-00857-t004:** ANCOVA results for PETQ of the experimental and control groups.

Variation	Source	Type III SS	df	MS	F	*p*	η^2^
CC	Group	2.22	1	2.22	39.27 *	0.00	0.30
	Error	5.14	91	0.057			
TS	Group	1.48	1	1.48	20.91 *	0.00	0.19
	Error	6.45	91	0.071			
CM	Group	1.05	1	1.05	16.44 *	0.00	0.15
	Error	5.78	91	0.064			
LA	Group	1.88	1	1.88	121.35 *	0.00	0.57
	Error	1.41	91	0.016			
PETQ	Group	3.09	1	3.09	37.33 *	0.00	0.29
	Error	7.54	91	0.083			

* *p* < 0.05.

## Data Availability

Data are available from the corresponding author upon reasonable request.

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
