# Peer review of "From Tradition to Innovation: Analyzing Strategies and Support for Enhancing Badminton Course Teaching Quality through Educational Technology"

_behavsci, 2024, doi:10.3390/bs14090857_

Round 1

Reviewer 1 Report

Comments and Suggestions for Authors

1. What is PE? You should explain within the text as physical education PE. The same as MAT

2. Regarding methodology, you said that it was a quasi-experimental design, but you described it as a random selection. What is important for quasi-experimental design?

3. What was the eligibility criteria for two groups?

4. What was the intervention course? Which activities were integrated?

5. Is Curriculum Development part related to the intervention? Can you combine it with intervention?

Comments on the Quality of English Language

Minor editing is required

Author Response

Responses to the comments of Reviewer #1

1) What is PE? You should explain within the text as physical education PE. The same as MAT

Response: Thank you for the reviewer's suggestion, the PE and MAT have been explained within the text.

2)Regarding methodology, you said that it was a quasi-experimental design, but you described it as a random selection. What is important for quasi-experimental design?

Response: Thank you for the reviewer's suggestion. Quasi-experimental design is a research methodology that aims to evaluate the effect of an intervention or treatment without the random assignment of participants to groups. While it lacks the randomization of true experimental designs, it can still provide valuable insights, especially in real-world settings where random assignment is not feasible. The randomization mentioned in our text refers to the selection and allocation of two classes for the teaching experiment. Based on the reviewer's suggestion, we have revised the wording to address this oversight.

3)What was the eligibility criteria for two groups?

Response: To ensure that the two groups are similar in terms of experimental factors, we recruited and screened students based on inclusion and exclusion criteria. The inclusion criteria were enrollment in a first-year badminton course and an age range of 18-20 years. Students who were admitted through the athletic talent program were excluded.

4) What was the intervention course? Which activities were integrated?

Response: Thank you for the reviewer's suggestion. Based on the team's developed materials covering aspects such as court equipment, game rules, and scoring methods Q&A (each unit approximately 4 minutes), each week after oral instruction, students were immediately asked to use their mobile phones for video-assisted teaching. We will make the suggested revisions and add explanations in the text.

5)Is Curriculum Development part related to the intervention? Can you combine it with intervention?

Response: Thank you for the reviewer's suggestion. The curriculum development in this study involves designing and implementing audio-visual intervention strategies as part of the intervention measures. The curriculum is designed based on the school's teaching syllabus to meet the learning objectives and needs of the badminton course. The intervention measures are adapted to the characteristics of audio-visual teaching, with the aim of improving learning outcomes. During the intervention process of the study, we integrated multimedia-assisted teaching to enhance students' learning outcomes and address issues that we had previously identified in our teaching.

Reviewer 2 Report

Comments and Suggestions for Authors

Thank you for the opportunity to review this manuscript. This work makes a unique and valuable contribution to the field of Physical Education and educational technology. Please consider the points below:

Introduction

- Page 1, lines 35-37: why must teachers employ a multifaceted approach to utilise various multimedia resources in creating real-time and interactive learning environments? Are you sure this is the only reason to ensure that the educational setting meets students' needs for maintaining an active attitude and high intrinsic motivation?

- Page 1, line 41: "Educators should cater to these differences as both teachers and learners" - what differences?

- Page1/2, lines 44-45: how can multimedia significantly improve the learning experience? Please explain. Consider making the remaining sentences of this paragraph into a new paragraph.

- Page 3, lines 101-103: online and distance teaching are both used in this sentence. I encourage you to be consistent with terminology and consider providing a definition.

Methods

- 2.1 Study Sample: is there any reason why gender split of participants is not identified?

- 2.2 Experimental Procedures: please explain what is meant by limitations in the educational environment?

- Why was a four-week time period selected? 

- What was it that the participants consented to exactly?

- 2.6 Research Limitations: please provide more information about why not all badminton activities were incorporated. What impact could this limitation have on the study overall?

Discussion

Page 7, line 244: this should state "highlights".

Page 7, lines 248-250: it is a big statement that online provision can achieve a quality of teaching comparable to offline classes. I encourage you to re-consider this statement, as you have examined badminton; there are many other pedagogical approaches and topics that you have not investigated. 

- Page 8, lines 253-254: if multimedia possesses unique qualities that traditional teaching methods cannot replace, then please provide examples. Consider also that traditional methods possess unique qualities that multimedia cannot replace too.

- Page 8, line 266: why is birth rate mentioned? What relevance does this have?

- Page 8, line 270: what is meant by attract and meet student needs? What needs?

- Page 8, lines 277-278: your statement that integrating MAT into PE can address these limitations is somewhat bold. Perhaps integrating MAT can facilitate more effective use of limited time and large student numbers without hindering student learning.

- Pages 8/9, paragraph lines 281-304: I encourage you to be careful in making claims based on self-report methods (PETQ scale); these have limitations and restrictions as quality and robust data collection instruments.

- Page 9, lines 305-322: How do you know that some of the claims you make about MAT do not occur with traditional teaching methods?

- Page 9/10, lines 323-340: there are several good points made in this paragraph, but including some of this information in the Introduction would be advantageous. For example, "physical education teachers often face common barriers when using technology" such as .....

Comments on the Quality of English Language

Thank you for the opportunity to review this manuscript. This work makes a unique and valuable contribution to the field of Physical Education and educational technology. Please consider the points below:

Introduction

- Page 1, lines 35-37: why must teachers employ a multifaceted approach to utilise various multimedia resources in creating real-time and interactive learning environments? Are you sure this is the only reason to ensure that the educational setting meets students' needs for maintaining an active attitude and high intrinsic motivation?

- Page 1, line 41: "Educators should cater to these differences as both teachers and learners" - what differences?

- Page1/2, lines 44-45: how can multimedia significantly improve the learning experience? Please explain. Consider making the remaining sentences of this paragraph into a new paragraph.

- Page 3, lines 101-103: online and distance teaching are both used in this sentence. I encourage you to be consistent with terminology and consider providing a definition.

Methods

- 2.1 Study Sample: is there any reason why gender split of participants is not identified?

- 2.2 Experimental Procedures: please explain what is meant by limitations in the educational environment?

- Why was a four-week time period selected? 

- What was it that the participants consented to exactly?

- 2.6 Research Limitations: please provide more information about why not all badminton activities were incorporated. What impact could this limitation have on the study overall?

Discussion

Page 7, line 244: this should state "highlights".

Page 7, lines 248-250: it is a big statement that online provision can achieve a quality of teaching comparable to offline classes. I encourage you to re-consider this statement, as you have examined badminton; there are many other pedagogical approaches and topics that you have not investigated. 

- Page 8, lines 253-254: if multimedia possesses unique qualities that traditional teaching methods cannot replace, then please provide examples. Consider also that traditional methods possess unique qualities that multimedia cannot replace too.

- Page 8, line 266: why is birth rate mentioned? What relevance does this have?

- Page 8, line 270: what is meant by attract and meet student needs? What needs?

- Page 8, lines 277-278: your statement that integrating MAT into PE can address these limitations is somewhat bold. Perhaps integrating MAT can facilitate more effective use of limited time and large student numbers without hindering student learning.

- Pages 8/9, paragraph lines 281-304: I encourage you to be careful in making claims based on self-report methods (PETQ scale); these have limitations and restrictions as quality and robust data collection instruments.

- Page 9, lines 305-322: How do you know that some of the claims you make about MAT do not occur with traditional teaching methods?

- Page 9/10, lines 323-340: there are several good points made in this paragraph, but including some of this information in the Introduction would be advantageous. For example, "physical education teachers often face common barriers when using technology" such as .....

Author Response

Responses to the comments of Reviewer #2

1) Introduction/ Page 1, lines 35-37: why must teachers employ a multifaceted approach to utilise various multimedia resources in creating real-time and interactive learning environments? Are you sure this is the only reason to ensure that the educational setting meets students' needs for maintaining an active attitude and high intrinsic motivation?

Response: Thank you to the reviewers for your suggestions and reminders. While it is indeed a key reason that teachers adopt a multifaceted approach to utilizing various multimedia resources to create real-time and interactive learning environments, it is not the only reason. We have revised the sentence to: "Teachers can leverage the advantages of multimedia by adopting a multifaceted approach to utilizing various multimedia resources to create real-time and interactive learning environments."

2) Page 1, line 41: "Educators should cater to these differences as both teachers and learners" - what differences?

Response: Thank you to the reviewers for your correction. The 'differences' in the phrase 'educators should cater to these differences' refer to the various types of multimedia used in the educational process, including symbols, images, pictures, audio, video, and animations. Students may have different preferences or strengths when engaging with these different media formats during learning. Educators should adjust their teaching methods accordingly to ensure that both teachers and learners can achieve the common goal of understanding and retaining the material. We have rewritten the paragraph to make it clearer.

3)Page1/2, lines 44-45: how can multimedia significantly improve the learning experience? Please explain. Consider making the remaining sentences of this paragraph into a new paragraph.

Response: Thank you to the reviewers for your suggestions and insights. Multimedia-assisted teaching can improve the learning experience through various advantages, such as enhancing understanding and memory, increasing learning interest and motivation, promoting multi-sensory learning, improving interactivity and engagement, and supporting differentiated learning. This is mainly because it combines multiple forms of media, including text, images, audio, and video, making the learning process more rich and dynamic, while helping students understand and grasp knowledge more efficiently and deeply. Based on the reviewers' suggestions, we have restructured the remaining sentences of this section into a new paragraph.

4)Page 3, lines 101-103: online and distance teaching are both used in this sentence. I encourage you to be consistent with terminology and consider providing a definition.

Response: Thanks for the reviewer's suggestion, corrected.

5)Methods/ 2.1 Study Sample: is there any reason why gender split of participants is not identified?

Response: Thank you to the review committee for your guidance. Based on the revisions, we have added gender and the number of participants to the Study Sample section and Table 1.

6)2.2 Experimental Procedures: please explain what is meant by limitations in the educational environment?

Response: The "limitations in the educational environment" refer to class size and student-teacher ratios (the number of students enrolled in each class and the gender ratio are uncontrollable factors), environmental and contextual factors (such as the time of day for classes—morning or afternoon—and whether air conditioning or lighting is available during lessons), as well as external influences (family background, socio-economic status, etc., which can limit students' learning ability or engagement, thus impacting educational outcomes).

7)Why was a four-week time period selected?

Response: The choice of a 4-week duration for this experiment is based on the research design and supported by relevant literature. The study plan was established according to the school's curriculum, where the badminton unit is set for an 8-week period. To avoid affecting the instruction of skills, our research planning meeting incorporated some foundational literature and recommendations. Rosenshine (2012) suggested considering the actual operational time of the school semester, while also taking into account the students' learning cycles. The purpose of short-term studies is to verify the immediate impact of specific teaching methods (Oakhill, Cain, & Elbro, 2014), and short-term studies often have a greater impact than long-term studies (Cheung & Slavin, 2013). Therefore, without interfering with skill and team-based learning, we referenced the 4-week period to test the effectiveness of multimedia video-assisted teaching.

Rosenshine, B. (2012). Principles of instruction: Research-based strategies that all teachers should know. American educator, 36(1), 12.

Oakhill, J., Cain, K., & Elbro, C. (2014). Understanding and teaching reading comprehension: A handbook: Routledge.

Cheung, A. C. K., & Slavin, R. E. (2013). The effectiveness of educational technology applications for enhancing mathematics achievement in K-12 classrooms: A meta-analysis. Educational Research Review, 9, 88-113.

8)What was it that the participants consented to exactly?

Response: before the course, we informed the students about the conditions and limitations of participating in the study, the research methods and procedures, and that participants could withdraw from the experiment at any time. All those who agreed to participate signed an informed consent form before the experiment started.

9)Research Limitations: please provide more information about why not all badminton activities were incorporated. What impact could this limitation have on the study overall?

Response: Thank you to the reviewers for your suggestions and insights. This study only selected four instructional units (court equipment, game rules, scoring methods, and Q&A). Therefore, the research limitation refers to the fact that not all badminton teaching activities were fully included. It means that other basic badminton movements or skills, such as wind-up, strike/hit/bat/stroke/shot, return, overhand stroke, underhand stroke, transitional ball, hair-pin shot, sliding step, cross step, etc., were not included.

10)Discussion/Page 7, line 244: this should state "highlights".

Response: Thanks for the reviewer's suggestion, corrected.

11)Page 7, lines 248-250: it is a big statement that online provision can achieve a quality of teaching comparable to offline classes. I encourage you to re-consider this statement, as you have examined badminton; there are many other pedagogical approaches and topics that you have not investigated.

Response: Thanks for the reviewer's suggestion, corrected.

12)Page 8, lines 253-254: if multimedia possesses unique qualities that traditional teaching methods cannot replace, then please provide examples. Consider also that traditional methods possess unique qualities that multimedia cannot replace too.

Response: Thank you for the suggestions and reminders from the review committee. Multimedia has some unique qualities that traditional teaching methods cannot replace, such as visual presentation, instant interaction, and overcoming geographical and time limitations. However, it should not be overlooked that traditional teaching methods also possess unique qualities that multimedia cannot replace, such as real-time interaction and emotional exchange between teachers and students, as well as group cooperation and in-person teaching activities. These are aspects that multimedia-assisted teaching finds difficult to fully replicate. Therefore, both have their irreplaceable features and should be flexibly applied based on teaching objectives and student needs. We will revise the content to balance the strengths and weaknesses of both teaching strategies.

13)Page 8, line 266: why is birth rate mentioned? What relevance does this have?

Response: This paragraph mentions the birth rate because, as Taiwan's birth rate declines (with the CIA's 2024 global fertility rate prediction placing Taiwan at the bottom of the list, ranked 227th with a fertility rate of 1.11), many universities, especially private ones, are facing the challenge of decreasing freshman enrollment, making admissions more difficult. To cope with this trend, many private schools have had to compress or reduce general education courses. However, as parents become increasingly concerned about the quality of education their children receive, expectations and demands for school performance and educational quality have risen as a factor in their choice of school. Therefore, schools aim to enhance the quality of instruction in hopes of attracting and retaining students in this highly competitive environment. If the review committee believes that this explanation is not highly relevant to the article, we can proceed with its removal.

14)Page 8, line 270: what is meant by attract and meet student needs? What needs?

Response: Thank you to the reviewers for your suggestions and insights. In this sentence, 'attract and meet student needs' refers to how PE teaching units should design and provide physical education programs based on students' interests, preferences, and expectations to ensure active participation in PE classes and meet their learning and developmental needs. Therefore, PE teaching units need to enhance the quality of instruction to attract student participation and address these needs. We have added an explanation in the text to clarify.

15)Page 8, lines 277-278: your statement that integrating MAT into PE can address these limitations is somewhat bold. Perhaps integrating MAT can facilitate more effective use of limited time and large student numbers without hindering student learning.

Response: Thank you for the suggestions and reminders from the review committee. We have revised the sentence as follows: Integrating MAT into PE presents an opportunity to more effectively address these issues without hindering students' learning. In other words, combining video materials with lesson plans can effectively guide students in mastering key movements and techniques, thereby improving teaching quality and enhancing students' learning outcomes.

16)Pages 8/9, paragraph lines 281-304: I encourage you to be careful in making claims based on self-report methods (PETQ scale); these have limitations and restrictions as quality and robust data collection instruments.

Response: Thank you for the suggestions and reminders from the review committee. We have revised the paragraph.

17)Page 9, lines 305-322: How do you know that some of the claims you make about MAT do not occur with traditional teaching methods?

Response: Thank you for the suggestions and reminders from the review committee. Many of MAT’s advantages stem from its ability to integrate multimedia elements into teaching, providing richer learning resources and interactive experiences. However, as you suggested, this does not mean that these effects cannot be achieved through traditional teaching methods (TT). In fact, some TT methods can effectively capture students' attention, especially when the teacher excels in verbal expression and technical demonstrations. MAT’s strength lies in its multimedia presentation and interactive features, but traditional teaching may also achieve similar results in different contexts. Therefore, we cannot entirely dismiss the possibility that TT may produce similar educational outcomes in certain situations. We will revise this paragraph accordingly.

18)Page 9/10, lines 323-340: there are several good points made in this paragraph, but including some of this information in the Introduction would be advantageous. For example, "physical education teachers often face common barriers when using technology" such as .....

Response: Thank you for the suggestions and reminders from the review committee. We have revised and added some of the information you suggested in the introduction.

Reviewer 3 Report

Comments and Suggestions for Authors

First of all, thank you for the opportunity to review this work. My suggestions for authors refer to:

1. The two hypotheses expressed (lines 124-127) are very similar and could be considered as approaching the same concept from slightly different perspectives. Both hypotheses refer to the improvement of the quality of instruction in physical education by using multimedia-assisted teaching compared to traditional teaching. Both hypotheses seem to be oriented towards the idea that multimedia-assisted teaching is superior to traditional teaching in terms of the quality of physical education instruction, so it might be redundant to separate them. Instead of having two very similar hypotheses, it would be more effective to formulate one comprehensive hypothesis that integrates both aspects.

2. Subchapter 2.6 I think it would be useful to be moved to the end of the Discussions chapter and of course to add other possible limits of the current research.

Author Response

Responses to the comments of Reviewer #3

1) The two hypotheses expressed (lines 124-127) are very similar and could be considered as approaching the same concept from slightly different perspectives. Both hypotheses refer to the improvement of the quality of instruction in physical education by using multimedia-assisted teaching compared to traditional teaching. Both hypotheses seem to be oriented towards the idea that multimedia-assisted teaching is superior to traditional teaching in terms of the quality of physical education instruction, so it might be redundant to separate them. Instead of having two very similar hypotheses, it would be more effective to formulate one comprehensive hypothesis that integrates both aspects.

Response: Thank you to the review committee for your guidance. Based on your suggestions, we have integrated the elements of the two similar hypotheses.

2) Subchapter 2.6 I think it would be useful to be moved to the end of the Discussions chapter and of course to add other possible limits of the current research.

Response: Thanks for the reviewer's suggestion, corrected.

Reviewer 4 Report

Comments and Suggestions for Authors

A very interesting article on new technological possibilities in physical education classes. However, having analysed the article, it must be concluded that in this form the article is not suitable for publication. 

The abstract lacks detailed information about the practical application of the research. 

Introduction. Analysing the introduction, one has to conclude that the authors wrote about many things: a little bit about PE, a little bit about cognitive skills, and at the end they presented the division and types of teaching. The introduction should be an introduction to the research conducted - unfortunately, this is not evident after reading the introduction. Also, very often the authors do not cite the things they write about - no citations in the introduction. The introduction section is completely amendable.

In the methods section - the authors do not write whether they studied men or women.

When presenting graphics, the authors do not write what the source of the photos was. 

On the other hand, in the first sentences of the discussion, the authors presented the aim of the study and the results obtained in relation to other scientific publications which was very good (line 239-253). However, in the second part of the discussion, the authors make very little reference or comparison of the obtained research results with publications from around the world. 

In the conclusion section - there is no detailed practical conclusion derived from the research. 

The authors should check the following scientific publications in the literature: 1, 13,15, 20, 23, 24, 26, 33, 36, 39) no pages.

Author Response

Responses to the comments of Reviewer #4

1) The abstract lacks detailed information about the practical application of the research.

Response: Thank you for the reviewer's suggestions. We have strengthened the abstract by adding detailed information about the practical application.

2)Introduction. Analysing the introduction, one has to conclude that the authors wrote about many things: a little bit about PE, a little bit about cognitive skills, and at the end they presented the division and types of teaching. The introduction should be an introduction to the research conducted - unfortunately, this is not evident after reading the introduction. Also, very often the authors do not cite the things they write about - no citations in the introduction. The introduction section is completely amendable.

Response: Thank you for the reviewers' suggestions. We have revised and strengthened the introduction to better present the research being conducted.

3)In the methods section - the authors do not write whether they studied men or women.

Response: Thank you to the review committee for your guidance. Based on the revisions, we have added gender and the number of participants to the Study Sample section and Table 1.

4)When presenting graphics, the authors do not write what the source of the photos was.

Response: Thank you for the reviewer's suggestion, the source has been added to the photos.

5)On the other hand, in the first sentences of the discussion, the authors presented the aim of the study and the results obtained in relation to other scientific publications which was very good (line 239-253). However, in the second part of the discussion, the authors make very little reference or comparison of the obtained research results with publications from around the world.

Response: Thank you for the reviewer's suggestion, corrected.

6)In the conclusion section - there is no detailed practical conclusion derived from the research.

Response: Thank you for the reviewer’s suggestions. We have added detailed practical conclusions.

7)The authors should check the following scientific publications in the literature: 1, 13,15, 20, 23, 24, 26, 33, 36, 39) no pages.

Response: Thank you for the reviewer's suggestion, the pages have been added to the text.

Round 2

Reviewer 2 Report

Comments and Suggestions for Authors

Thank you for addressing the reviewer feedback provided, this has made substantial improvements to the manuscript. I am supportive of the manuscript being published but ask that the points below are considered:

- Page 2, line 46: the statement about "Given the challenges of teaching complex skills in PE" - what challenges are you referring to? Examples need to be included. Why do challenges exist?

- Page 2, line 56: "can be better explained by multimedia tools" - how do you know this? What is this statement based on? Can be better explained by multimedia tools as opposed to what?

- Page 2, line 65: currently states "stendents:, should state "students".

- Page 2, line 68: .PE - should be a space between . and PE.

- Page 2, 74: what is reasonable use of multimedia teaching in PE classes? What is meant by reasonable?

Comments on the Quality of English Language

Quality of English is sound, but ask that the language and grammar around the points of feedback I have provided is reviewed.

Author Response

Responses to the comments of Reviewer #2

1) Page 2, line 46: the statement about "Given the challenges of teaching complex skills in PE" - what challenges are you referring to? Examples need to be included. Why do challenges exist?

Response: Thank you for the valuable feedback from the reviewers. Teaching complex motor skills in physical education indeed presents numerous challenges. These difficulties include:

Technical challenges: precise movements and correct body positioning, such as the serving motion in badminton.

Coordination challenges: the swing motion in badminton, which requires coordinated arm and wrist movements.

Comprehension difficulties: understanding the court layout or the rules.

For students, it is challenging to understand and replicate these complex movements through verbal instructions or demonstrations alone. Therefore, utilizing multimedia technology can effectively address these issues. For example, video demonstrations can help students understand and master complex motor skills more intuitively.

2)Page 2, line 56: "can be better explained by multimedia tools" - how do you know this? What is this statement based on? Can be better explained by multimedia tools as opposed to what?

Response: Thank you for the valuable feedback from the reviewers. We have added research support. According to Dou's findings, multimedia tools provide both visual and auditory stimuli, which help students better understand and retain complex content. Additionally, in our practical observations, we found that using multimedia tools (such as instructional videos and animations) allows for a more intuitive demonstration of badminton techniques and game strategies, significantly enhancing student learning outcomes.

In response to "as opposed to what," our comparison refers to traditional teacher-centered oral instruction methods. These methods often fall short in explaining complex technical movements and game strategies, whereas multimedia tools offer more intuitive and detailed presentations, thereby improving student comprehension and performance. Following your suggestion, we will include this explanation in the manuscript.

Dou, Y. Badminton Teaching Mode in Network Teaching Platform under Multimedia Environment. Int. J. Web-Based Learn. Teach. Technol. 2023, 18 (2), 1-18.

3)Page 2, line 65: currently states "stendents:, should state "students".

Response: Based on the reviewers' suggestions, we have revised the manuscript accordingly.

4)Page 2, line 68: .PE - should be a space between . and PE

Response: Based on the reviewers' suggestions, we have revised the manuscript accordingly.

5) Page 2, 74: what is reasonable use of multimedia teaching in PE classes? What is meant by reasonable?

Response: Thank you for the insightful comments. we define reasonable use as the appropriate integration of multimedia resources to supplement and enhance traditional teaching methods, rather than completely replacing them. Reasonable use should align with the teaching objectives and be implemented in a way that maximizes the effectiveness of instruction. Specific practices that constitute reasonable use include:

Pre-class Preparation: Providing multimedia videos related to the motor skills to be taught, giving students a preliminary understanding before the actual class.

In-class Demonstrations: Using high-quality video demonstrations to break down complex movements, allowing students to see each detail more clearly than with verbal explanations alone.

Real-time Feedback: Recording students' movements during practice sessions and comparing them with standard movement videos. This helps students identify their mistakes and make improvements more intuitively.

Post-class Review: Offering multimedia resources for students to review and practice at home, helping to reinforce what they have learned in class.

The significance of reasonable use lies in avoiding over-reliance on technology while not neglecting the benefits of traditional teaching methods such as interpersonal interaction and immediate feedback. The aim is to combine both approaches to maximize teaching effectiveness. Following your suggestion, we will include an explanation in the manuscript.

Reviewer 4 Report

Comments and Suggestions for Authors

Many thanks to the authors of the scientific article for responding to the comments and suggestions sent. All comments have been corrected by the authors. This has led to an increase in the quality of the scientific paper. As a reviewer of the submitted article, I am satisfied with the corrections made. One remark should be changed. The authors, in signing the pictures, used a www. link from YouTube - this is not a scientific source. Furthermore, if such a source has been used - the authors must have permission from the owner of the uploaded images/videos on YouTube

Author Response

Responses to the comments of Reviewer #4

1) Many thanks to the authors of the scientific article for responding to the comments and suggestions sent. All comments have been corrected by the authors. This has led to an increase in the quality of the scientific paper. As a reviewer of the submitted article, I am satisfied with the corrections made. One remark should be changed. The authors, in signing the pictures, used a www. link from YouTube - this is not a scientific source. Furthermore, if such a source has been used - the authors must have permission from the owner of the uploaded images/videos on YouTube

Response: Thank you for your insightful comments and valuable suggestions. We have used a YouTube link in Figure 2. However, the linked video was created and uploaded by the authors themselves. The content's source and rights belong to the authors, ensuring that they possess full copyright and legal rights. This guarantees the credibility and legality of the materials. Following your suggestion, we will include an explanation in the manuscript.
